# SigPrimedNet: A Signaling-Informed Neural Network for scRNA-seq Annotation of Known and Unknown Cell Types

**DOI:** 10.3390/biology12040579

**Published:** 2023-04-10

**Authors:** Pelin Gundogdu, Inmaculada Alamo, Isabel A. Nepomuceno-Chamorro, Joaquin Dopazo, Carlos Loucera

**Affiliations:** 1Computational Medicine Platform, Andalusian Public Foundation Progress and Health-FPS, 41013 Sevilla, Spain; 2Computational Systems Medicine, Institute of Biomedicine of Seville (IBIS), Hospital Virgen del Rocio, 41013 Sevilla, Spain; 3Dpto. de Lenguajes y Sistemas Informáticos, Universidad de Sevilla, 41013 Seville, Spain; 4Bioinformatics in Rare Diseases (BiER), Centro de Investigación Biomédica en Red de Enfermedades Raras (CIBERER), FPS, Hospital Virgen del Rocío, 41013 Sevilla, Spain; 5FPS/ELIXIR-es, Hospital Virgen del Rocío, 42013 Sevilla, Spain

**Keywords:** scRNA-seq, deep learning, explainable artificial intelligence, cell signaling, cell-type identification

## Abstract

**Simple Summary:**

Single-cell data has enabled the study of cell dynamics at an unprecedented resolution. Cell type and functional annotation are crucial to address during any analysis involving transcriptomic data at the cell level since both annotations provide the basis to understand the complex biological processes behind the communication machinery. We propose SigPrimedNet, a data-driven solution to identify cells while learning a functional summarization of signaling measurements by incorporating the knowledge stored in pathway databases. To do so, we decompose each signaling pathway into canonical effector circuits, which act as a minimal functional unit. These circuits inform the design of a cell-type classification neural network model, which allows us to extract meaningful features that act as a proxy of the signaling activity of any given cell. Furthermore, we train an unsupervised anomaly detection algorithm on the inferred activities, which enables the model to identify unknown cells when working with previously unseen cells. To illustrate the performance of the proposed model we conduct a series of experiments over publicly available data with promising results across every task: cell-type annotation, unknown cell-type identification, and clustering. Finally, we showcase the biological richness of the signaling activity learned by the model.

**Abstract:**

Single-cell RNA sequencing is increasing our understanding of the behavior of complex tissues or organs, by providing unprecedented details on the complex cell type landscape at the level of individual cells. Cell type definition and functional annotation are key steps to understanding the molecular processes behind the underlying cellular communication machinery. However, the exponential growth of scRNA-seq data has made the task of manually annotating cells unfeasible, due not only to an unparalleled resolution of the technology but to an ever-increasing heterogeneity of the data. Many supervised and unsupervised methods have been proposed to automatically annotate cells. Supervised approaches for cell-type annotation outperform unsupervised methods except when new (unknown) cell types are present. Here, we introduce SigPrimedNet an artificial neural network approach that leverages (i) efficient training by means of a sparsity-inducing signaling circuits-informed layer, (ii) feature representation learning through supervised training, and (iii) unknown cell-type identification by fitting an anomaly detection method on the learned representation. We show that SigPrimedNet can efficiently annotate known cell types while keeping a low false-positive rate for unseen cells across a set of publicly available datasets. In addition, the learned representation acts as a proxy for signaling circuit activity measurements, which provide useful estimations of the cell functionalities.

## 1. Introduction

Recent high-throughput technology developments are transforming our view of complex biological systems by providing a detailed picture of their individual components. Single-cell RNA sequencing (scRNA-seq) has enabled RNA activity to be profiled in individual single cells by obtaining profiles of thousands of cells in heterogeneous environments [1]. scRNA-seq increases our understanding of the cell as a functional unit revealing new populations of cells with gene expression profiles previously unnoticed in conventional analyses of bulk cell populations [2].

Facing the huge amount of data provided by scRNA-seq technology, one of the major challenges is cell-type identification within a diverse population of sequenced cells. This challenge, also known as cell retrieval or cell-type annotation, consists of inferring the type of a given cell by querying a reference database of annotated scRNA-seq data. Unsupervised methods, such as clustering analysis, find the closest cell to a sample given a population of cells. However, single-cell data contains high levels of noise from heterogeneous sources, and to mitigate such problems, dimensionality reduction is usually performed before clustering. Scmap projection algorithm [3] explores different strategies for feature selection as highly variable genes (HVGs) [4] and genes with a higher number of dropouts (zero expression) than expected determined using M3Drop [5]. The most popular methods for dimensionality reduction are based on Principal Component Analysis (PCA) [6], dropout modeling (ZIFA) [7], t-distributed stochastic neighbor embedding (TSNE) [8] or uniform manifold approximation and projection (UMAP) [9]. Single reference mapping methods are growing in popularity as Seurat’s supervised principal component analysis [10], single-cell architecture surgery (scArches) [11], or an extension of Harmony [12] to map query datasets by minimal modification of the reference atlas [13]. However, the implicitly used latent dimensions for joint data representation are not directly interpretable, and it is a major drawback of these methods [14]. Currently, the development of interpretable models by the addition of statistical assumptions or prior biological information is a trend, but the former approaches have not yielded sufficiently useful latent spaces in the context of scRNA-seq analysis [14].

Supervised methods use a labeled reference to learn a function that maps transcriptomic profiles to cell types. Thereafter, new cells are annotated using the learned mapping. Model training (learning the map) is usually a time-consuming process due to the large size of the reference databases [3], while inference (applying the learned function) is faster and less laborious than the two-step process associated with unsupervised methods [15]. Furthermore, supervised training for cell-type annotation usually performs better than unsupervised methods in most datasets, although this is not the case when unknown cell types arise [16]. One of the more promising methods to overcome such limitations is SciBet, which uses a combination of statistical learning to find informative genes, a multinomial approximation for cell-type annotation, and building a synthetic reference cell to estimate out-of-distribution transcriptomic profiles. Scibet outperforms other state-of-the-art methods like Seurat v3 and scMap across several experiments, achieving a high prediction accuracy while keeping a low false-positive rate when annotating unseen cell types.

In this work, we present SigPrimedNet a domain-informed Artificial Neural Network (ANN) that overcomes the limitations associated with supervised learning methods by combining a signaling circuits-informed sparse architecture with an anomaly detection procedure that uses the latent structure learned by the ANN to elucidate if any given cell is of unknown origin. Sparse domain-informed neural networks are used to solve complex biological problems by incorporating domain-specific constraints on the underlying architectures to develop more interpretable models that avoid overfitting through regularization [17]. For example, P-NET [18] incorporates different biological entities to aid in decision-making when dealing with prostate cancer patients, whereas Dcell models gene interactions on cell growth in yeast [19]. In the context of cell annotation [20] uses algorithmically crafted clusters of protein-protein and protein-DNA interactions to provide the sparse structure, but cannot classify unknown cells.

Our previous work on cell-type identification [21] used broader, all-encompassing, pathways and lacked any form of out-of-distribution learning, which hampered its usefulness when the query dataset representation showed more heterogeneity in cell populations. Contrarily, SigPrimedNet offers a more fine-grained functional characterization of the cell populations due to the use of more specific effector-based signaling proxies based on recent developments in mechanistic models of cell signaling, which ultimately triggers cell functionality and dictates cell behavior and fate [22]. Our method outperforms Scibet, Seurat v3, and ScMap when dealing with unknown cell types while providing a comparable performance on tasks where no cells should be labeled as unknown (using the experiments proposed in [15]). To the best of our knowledge, SigPrimedNet is the first supervised Domain-informed Sparse Neural Network to incorporate unknown cell-type identification.

## 2. Materials and Methods

### 2.1. Datasets

In this manuscript, we use three publicly available datasets, which we have called PBMC, Immune, and Melanoma dataset to facilitate their reference throughout the manuscript. All of them are publicly available on two platforms, Gene Expression Omnibus (GEO [23]) and 10× Genomics [24], moreover, they are human sequencing data. The datasets used in this work have been obtained from [15] (PBMC and Melanoma) and [25] (Immune). See Table 1 for cell type details.

#### 2.1.1. PBMC Dataset

The full version of the fresh peripheral blood mononuclear cells (PBMCs) datasets is publicly available in 10x Genomics [24]. In this work, we use the preprocessed version proposed in [15], which consists of 2500 cells randomly sampled for each cell type: CD14+, CD19+, CD34+, CD56+, CD8+ Cytotoxic, CD4+/CD45RO+ Memory, and Treg cells. In addition, to test the reliability of the model with unbalanced datasets, we have randomly undersampled each cell type (using a proportion of 0.2, 0.4, and 0.6 of the original population) to produce a total of 21 synthetic datasets derived from the PBMC dataset.

#### 2.1.2. Immune Dataset

This dataset profiles the transcriptomes of bone marrow and peripheral blood-derived hematopoietic cells, which are publicly available from GEO database [23] with identifiers GSE137864 and GSE149938. The dataset profiles 7 cell types for 9456 samples (see Table 1) using a unique molecular identifier (UMI) counting [26]. To be more precise, CD34+ HSPCs, B cells, NK cells, T cells, monocytes, neutrophils, and erythrocytes for bone marrow, and together with regulatory B, naive B, memory B, cytotoxic NK, cytokine NK and T cells for peripheral blood-derived differentiated cells.

#### 2.1.3. Melanoma Dataset

This human melanoma scRNA-seq dataset has malignant cells, CD8+ and CD4+ T cells, B cells, natural killer (NK) cells, macrophages, cancer-associated fibroblasts (CAFs), and endothelial cell types. The cell types of CAF, malignant, and endothelial cells are combined in one group called *negative cell*. In [15] they propose a filtered version of the dataset, which profiles 6 cell types for 6173 samples (see Table 1). The dataset is split into two subsets called reference and query with 70–30% sampling size, where the negative cells only appear in the query set. Note that, contrary to Scibet, SigPrimedNet does not rely on an external synthetic reference cell constructed from the aggregation of several single-cell datasets, so we do not make use of the massive reference set described in [15].

### 2.2. Analysis Workflow

In this work, we propose an analysis workflow that tries to show how our proposed model (SigPrimedNet) can correctly identify previously unseen cell types without losing the advantages of supervised learning (fast and accurate known cell-type assignment) while providing a biologically useful latent space. The workflow (Figure 1) can be summarized in three steps: (A) data processing and architecture design, (B) knowledge extraction from learned representations (*interpretation*), and (C) cell-type inference.

In broad terms, the model works as follows: (i) the weights of the first hidden layer of a dense network are constrained by a binary matrix that encodes the biological information extracted from the Kyoto Encyclopedia of Genes and Genomes (KEGG) [27], (ii) any given training dataset is decomposed into two sets (learning and validation) stratified by cell-type, (iii) the model is fitted to the learning set while using the validation for early stopping the training, (iv) computes the learned representation (encoding) of the learning and validation sets by evaluating the activations of the last hidden layer, (v) fits an anomaly detection algorithm using the encoding of the learning set as the features, and, (vi) establishes a threshold for detecting anomalies (unknown cell types) using the validation encodings. When a new cell is evaluated, the model computes the corresponding encoding, decides if the cell is of an unknown cell type by applying the anomaly detection algorithm along with the learned threshold, and, finally if the cell is not an anomaly the cell-type mapping learned by the ANN is applied.

To check the performance of the model (when all cell types are known) we have followed [15] using the resampled PBMC dataset to conduct a 50 times repeated cell-type stratified cross-validation. Whereas, to test the capability to identify unknown cell types, we have followed the negative cell melanoma experiment as proposed by [15]. Finally, the functional interpretability of our model has been tested using the Immune dataset, where we have also checked the performance by means of 30 times repeated 10-fold cross-validation strategy (all cell types are known).

### 2.3. Model Design

The architecture of the SigPrimedNet is defined as a dense network (all nodes in any given layer are connected to all the nodes of the adjacent layers), where the input layer (one node for each gene) is connected to a signaling-informed layer (the first hidden layer), which is wired to a new dense layer (the encoding layer). Finally, a softmax layer (Equation 2) connects the network to the output (the cell types). The model uses Rectified Linear Units (ReLU) [28] activation functions (Equation 1) except for the output layer. To train the network, we use the categorical cross-entropy loss function (Equation 3), where each known cell type represents a category.
(1)relu(z)=max(0,z).
(2)$$\mathrm{softmax}\left(z_{i}\right)=\frac{e^{z_{i}}}{\sum_{j=1}^{n_{c}}e^{z_{j}}}.$$
(3)$$-\sum_{j=1}^{n_{c}}y_{i,j}\log\left(p_{i,j}\right)$$
where z refers to real-valued data, $n_{c}$ to the number of cell types, $y_{i,j}$ is 1 if cell type *j* is the correct classification for observation *i*, 0 otherwise. Finally, $p_{i,j}$ is the probability that the observation *i* belongs to cell type *j*.

### 2.4. Data Preprocessing

Count data is preprocessed using the Transcripts per Million (TPM) normalization method [29]. To preprocess unique molecular identifier (UMI) data we use Seurat v37 with default parameters (each cell UMI count is normalized using size-factor 10,000). In either case, we end with a gene-wise rescaling to $\left[-1,1\right]$ after a logarithmic transformation of the preprocessed data.

### 2.5. Signaling-Primed Sparsity-Inducing Layers

SigPrimedNet is an ANN informed by a set of signaling circuits extracted from KEGG. Each pathway is decomposed into multiple effector circuits, so-called because they are the subpathways that end in effector proteins, which are responsible for triggering the associated function. Each effector node (a node with no descendants) defines an effector circuit along with the nodes that lead to it. To parse KEGG and decompose the resulting pathways into effector circuits we have used the HiPathia R package (v 2.11.4) [22]: the resulting (human) pathway list has been curated to remove those related to specific diseases, which totals 92 pathways that give rise to 1210 circuits (see Table A1). Note that our implementation can be extended to other pathway databases as long as each signaling pathway can be decomposed into functional subpathways.

Therefore, given a signaling pathway P, its associated directed graph, and $\left\{g_{0},\ldots ,g_{n}\right\}$ the set of genes that belong to P, we build the indicator matrix for P as follows: (i) detect the pathway effector (nodes with no descendants, $\left\{e_{0},\ldots ,e_{m}\right\}$, and receptor (nodes with no ascendants) nodes, (ii) for each effector node *e*, define an effector circuit $C_{e}$ as the subgraph that contains all the receptor nodes, $\left\{r_{0}^{e},\ldots ,r_{k}^{e}\right\}$, that is connected to *e*, (iii) construct an indicator vector ${\vec{c}}_{e}$ where ${\vec{c}}_{e}\left(i\right)=1$ if $g_{i}\in C_{e}$, and ${\vec{c}}_{e}\left(i\right)=0$ otherwise. Then, the indicator matrix for pathway P is defined as $I\left(P\right)={\left[{\vec{c}}_{l}\right]}_{l=1}^{m}$. See Figure 2 for a simplified visual representation of how to build an indicator matrix.

To compose the signaling-informed layer each pathway is decomposed into its corresponding indicator matrix, which is used to build the indicator matrix $I_{S}$ that informs the signaling layer *S* by performing the outer join of the previous matrices. Trivially, $I_{S}$ is an indicator matrix with $I_{S}\left(i,j\right)=1$ if gene *i* belongs to circuit *j*, and $I_{S}\left(i,j\right)=0$ otherwise, where *i* and *j* traverse the set of all the signaling genes and circuits, respectively. This matrix informs the first hidden layer of the model: (i) the layer has as many nodes as effector circuits, (ii) the layer is initialized using Glorot uniform [30], and (iii) a weight that connects an input gene *i* to a node *j* is set to 0 if the corresponding entry in the indicator matrix is 0 (i.e. gene *i* does not belong to circuit *j*).

Therefore, the kernel $W_{S}$ of an informed layer *S* can be written as (Equation (Equation 4)):(4)$$W_{S}=W⊙I_{S}$$
where W is a $\left(n_{genes},n_{circuits}\right)$ real valued tensor, $I_{S}$ is the indicator matrix of dimension $\left(n_{genes},n_{circuits}\right)$, and ⊙ refers to element wise (Hadamard) product.

The integration of a signaling-informed layer into the ANN has two aims: on the one hand, the sparsity induced by the informed layer has a regularization effect that prevents overfitting [17,18,20], and on the other hand, the learned representation using effector circuits provides a useful representation of the data through the associated functions, which helps to mitigate the problems associated to uninformative latent spaces [14].

### 2.6. Network Training and Inference

To add another source of regularization as well as to provide the model with the ability to identify unknown cell types, we split each training set into learning and validation subsets. The model is fitted (using the ADAM optimizer [31]) in a fully supervised way (the cell types are the response) to the learning set using the validation for early stopping of the training phase. Thereafter, we encode both subsets using the resulting network. A Local Outlier Factor (LOF) [32] model is fitted using the learning encodings as the features, whereas the validation set is used for setting a threshold on the similarity score. With these artificial splits, we avoid the overconfidence associated with ANN when evaluating the data where it has been fitted [33], resulting in a more realistic threshold. The threshold is set to the mean of the *w* measure of the similarity score distributions across the cell types, where *w* represents the maximum allowed deviation from the distribution set as (Equation (Equation 5)):(5)$$w^{c}=q_{1}^{c}-1.5\left(q_{3}^{c}-q_{1}^{c}\right)$$
where *c* represents a cell type, and $q_{i}$ its *i*-th quartile.

To predict the cell type of a new sample, the model first encodes its preprocessed transcriptomic profile, then computes the similarity score associated, decides if it is of an unknown cell type (labeling as unassigned) based on the learned threshold and, if this is not the case the model annotates the cell using the mapping function learned during the supervised training.

Therefore, we exploit the richness of the representations learned by SigPrimedNet using an unsupervised anomaly detection algorithm (LOF), which locates unusual data points by evaluating each point’s local deviation from its neighbors. The LOF algorithm is based on the local density concept, in which locality is determined by K-nearest neighbors (KNN), whose distances are used for density-based scores. Finally, a point is considered an outlier if and only if the LOF score is greater than one. However, we compute a more realistic threshold by using the secondary (validation) set. See Figure 3 for a visual representation of the SigPrimedNet’s prediction mechanism.

See Table 2 and Table 3 for a summary of the design choices and the training and inference times, respectively.

### 2.7. Functional Proxies and Representation Learning

Once the model has been fitted to a collection of annotated cells, we extract the features learned by the ANN, also known as representation learning [34], by detaching the last layer and computing the activations of the encoding layer. As the model has learned to map the gene profiles to the cell types, the encoding layer captures a lower-dimensional representation of the data necessary for the mapping. Note that, if the activations are computed for the signaling-informed layer we obtain a functional representation of the data as the nodes act as a proxy for the effector circuits. For visualization purposes, we can map the encodings to a 2D space by using TSNE (See Appendix A).

## 3. Results and Discussion

We provide here a series of validation procedures to test the performance of SigPrimeNet under different scenarios: a synthetically balanced data set based on PBMC where all cell types are known, a synthetic collection of unbalanced data sets made by undersampling each of the cell types that appear in PBMC, a real-world unbalanced data set (Immune) where the cell types are known and a data set (Melanoma) built for benchmark unknown cell-type identification methods. In the Appendix A, we also provide results for a two-layer version of SigPrimedNet (adding a second dense hidden layer), and a set of experiments designed to showcase the supervised performance of SigPrimedNet with the aim of making it easier to compare it to other methods that lack the ability to identify unknown cell types.

### 3.1. Model Performance When All Cell Types Are Known

#### 3.1.1. Synthetically Balanced PBMC

We tested the performance of our method employing 50 times repeated stratified by cell type 10-fold cross-validation schema using the *balanced* PBMC dataset (see Materials). The confusion matrix aggregated across the test folds shows that SigPrimedNet has a high ability to distinguish between cell types, as can be seen in Figure 4. In general, SigPrimedNet exhibits excellent performance across all the cell types with a slight decrease when dealing with those that are very closely related, such as Memory and Regulatory T cells. It should be noted that these results are similar to those obtained by Scibet and better than those obtained by Seurat and Scmap in a similar experiment shown in [15] since our approach reduces the misclassification of cytotoxic T cells.

#### 3.1.2. Synthetically Unbalanced PBMC

To check the model performance in unbalanced scenarios, while still holding some control over the cell populations, we have randomly undersampled one cell type at a time in the PBMC dataset for different undersampling ratios (0.2, 0.4, 0.6). Then, we evaluated the performance of SigPrimedNet using a 10-fold cross-validation schema for each of the simulated datasets.

Figure 5 shows the aggregated cross-validation matrix for each synthetic dataset. As expected, underpopulated cell types lead to a decrease in the predictive power with respect to the minority class for those cell types that were hard to classify originally (Treg, Memory), while the performance of cytotoxic T cells is severely hampered when undersampling their population, similar to the experiments conducted with the balanced PBMC dataset in [15]. However, the performance of the model over the other known types remains at levels equivalent to those obtained when evaluating SigPrimedNet in the balanced scenario. In addition, the rate of cells incorrectly labeled as unassigned remains in the same range as in the *balanced* simulation.

#### 3.1.3. Real-World Unbalanced Scenario

To check the performance of our model in a class-imbalanced scenario we performed 30 times cell-type stratified repeated 10-fold cross-validation using the Inmune dataset. Despite the added difficulty due to the disproportion between the classes, a ratio of 7.72 between the highest (HSPCs) and lowest (T cells) populated cell types, our method could still provide a high discriminating power as can be observed in the aggregated confusion matrix depicted in Figure 6. Most misclassifications are cells incorrectly labeled as HSPCs, which could be explained in machine learning terms, as a bias towards the majority class (HSPCs), or in biological terms, since HSPCs are very heterogeneous with transcriptomic profiles that match other cell types patterns [35,36]. Furthermore, the proportion of incorrectly labeled cells remains low as shown in the PBMC experiments.

#### 3.1.4. Design Comparison

The expressive power of the informed layer is evident when comparing the results of the two designs tested in this paper: the model’s performance is not noticeably improved by adding more capacity to the network by including a dense layer. Thus, the signaling-informed layer is capable of constructing, by itself, the necessary meta-features to differentiate cell types from the point of view of cell signaling. This can be deducted by inspecting Figure 7: the recall and the proportion of cells with an assigned label are higher in the one-layer design, while the precision is similar for both designs. Note that we have used the weighted version of precision and recall to account for the label imbalance. See Appendix A for the complete set of results for the two-layer design.

### 3.2. Unknown Cell-Type Identification

#### Novelty Detection in the Melanoma Dataset

Due to the incomplete nature of the reference scRNA-seq data, cell types not present in the reference dataset may be falsely predicted as those used during the model training. To analyze our approach to this issue, we used the Melanoma dataset with immune cells as positive cells and the other cells as negative cells. Figure 8 depicts the confusion matrix for the case study of false-positive control, with normalization for each row (origin label): the task consists in annotating the negative cells as unassigned while assigning the corresponding label to the other cells. Note that negative cells including malignant cells, CAF cells, and endothelial cells were removed from the training set. Query cells identified as anomalies by SigPrimedNet were labeled as unassigned. The results show that our method consistently outperforms Scibet for all the known labels (except NK) while maintaining a similar false-positive ratio. As mentioned in the Datasets section, this experiment was designed for this specific task in [15], and we have been able to reproduce it with both models: Scibet and SigPrimedNet. This NK deficit could be easily understood as it is something shared across all the experiments conducted: very low-populated cell types are harder to classify.

The Melanoma dataset was used in our previous study with a limited pathway-driven neural network (PDNN) [21], which only works for supervised tasks. The performance is similar between PDNN and SigPrimedNet with balanced accuracy scores of 0.844 and 0.8837 for the test split, respectively (see Table 4 for a more comprehensive comparison). Note that the results are not fully comparable since the dataset was filtered in [21] by removing all the negative cells in order to be able to use the PDNN (which results in a more favorable scenario for supervised models, like PDNN). Note that if the PDNN is used when unknown cells are present, it would label all the unknown cells with one of the known labels (a critical limitation), which can be assessed by looking at the PDNN (*) entry in Table 4 where we have run the PDNN model on the full dataset. This is not the case for SigPrimedNet, where unknown cell types are properly labeled as “unassigned”.

Figure 9 shows the distribution of similarity scores. The graph shows the similarity scores computed by fitting the ANN to the learning set (70% of the reference), then we fit a LOF model using their encodings as features, and finally, a threshold is learned to use the similarity scores of the remaining 30% validation (blue colored). When assigning labels to the test (query unseen cells), the first step is computing the ANN encodings followed by the LOF scores (ocher colored): those scores below the threshold are labeled as unassigned, and the remaining cells are assigned the cell type using the mapping learned by SigPrimedNet. Appendix A shows an analogous result for the two-layer architecture, although the performance is worse than the one-layer interpretable design used here.

### 3.3. SigPrimedNet Provides Biologically Interpretable Results

To illustrate the potential of our approach in producing biologically interpretable results, we have selected, for each cell type, the ten highest-weighted nodes (Table A2) from the signaling-informed layer, each representing a circuit from KEGG (Table A1).

For example, circuit Hedgehog signaling pathway (hsa04340): GLI SUFU is known to be involved in the control of hematopoietic differentiation [37], and it is present in the rank for HSPCs, NKs, erythrocytes, and B cells. The GO annotations for this circuit include cell differentiation and cell proliferation.

Another example is circuit Hippo signaling pathway (hsa04390): SERPINE1, which is present in ranks for B cells, NKs and HSPCs. Ref. [38] describe the role of SERPINE1 in the regulation of immune-related biological processes in glioma, relating high expression of SERPINE1 to gene expression patterns enriched in immune-related signaling pathways such as B cell receptor signaling pathway, Natural Killer cell mediated cytotoxicity, primary immunodeficiency, and T cell receptor signaling pathway, among others. Additionally, ref. [39] describe a mechanism by which PAI-1 (SERPINE1) regulates the localization of HSPCs between the bone marrow or its migration to other tissues. HSPCs also list circuits regulating cell survival and cell adhesion, relevant for their proliferative activity as hematopoietic cell precursors.

We also find that the circuit Calcium signaling pathway (hsa04020): Sphingosine 1-phosphate is active for monocytes and neutrophils. Ref. [40] provide evidence for the need for SphK2 kinase in processes of intracellular catalytic lipid degradation, which should be necessary for the phagocytic activity of monocytes and neutrophils.

Neutrophils also list circuits related to secretion and cellular mobility. Interestingly, among the neutrophils rank we find a circuit from the Melanogenesis pathway (hsa04916): DCT, that is implied in tyrosine metabolism. Neutrophils release several types of amino acids upon adhesion and spreading onto fibronectin, a process especially relevant in tissues undergoing healing and regeneration processes [41]. This connects to the already described link of neutrophilic activity to inflammation-related skin pigmentation [42].

The rank for Erythrocytes’ top 10 includes circuits related to the regulation of the location of precursor cells during hematopoiesis, regulation of apoptosis, and transendothelial migration. It also includes circuits that could regulate erythrocyte micromechanical properties and fluidity, which are necessary to adapt the size and viscosity enabling circulation through thin terminal capillaries [43].

The top-ranked circuits for T cells include the circuit Cell cycle: TFDP1 E2F4, which regulates a circuit regulating the entry of cells in the S-phase of the cell cycle. Although T cell selection occurs in the thymus, there is evidence that they undergo further differentiation in peripheral tissues [44].

## 4. Conclusions

SigPrimedNet is a highly efficient neural network for cell-type annotation in single-cell transcriptomics, one of the main challenges arising from a field with exponential growth, while providing useful biological features. The tool has been successfully tested on three tasks, namely: supervised cell type classification, unknown cell type annotation, and representation learning usefulness. To that effect, different publicly available benchmarks on multiple datasets have been carried out, with an outstanding known cell-type annotation performance while keeping a low false-positive rate for cell types unknown to the model. The ability to successfully identify cells of unknown origin lies in the high expressiveness of the features learned by the neural network, which are successfully used to train an unsupervised secondary model that detects anomalous cell types. These features are proxies for the signaling circuits used to inform the layers of the model, which have a regularization benefit due to weight sparsification and present a meaningful set of biological functions. The model has very low latency when annotating new cells and provides rich and useful interpretable features.

## Figures and Tables

**Figure 1 biology-12-00579-f001:**
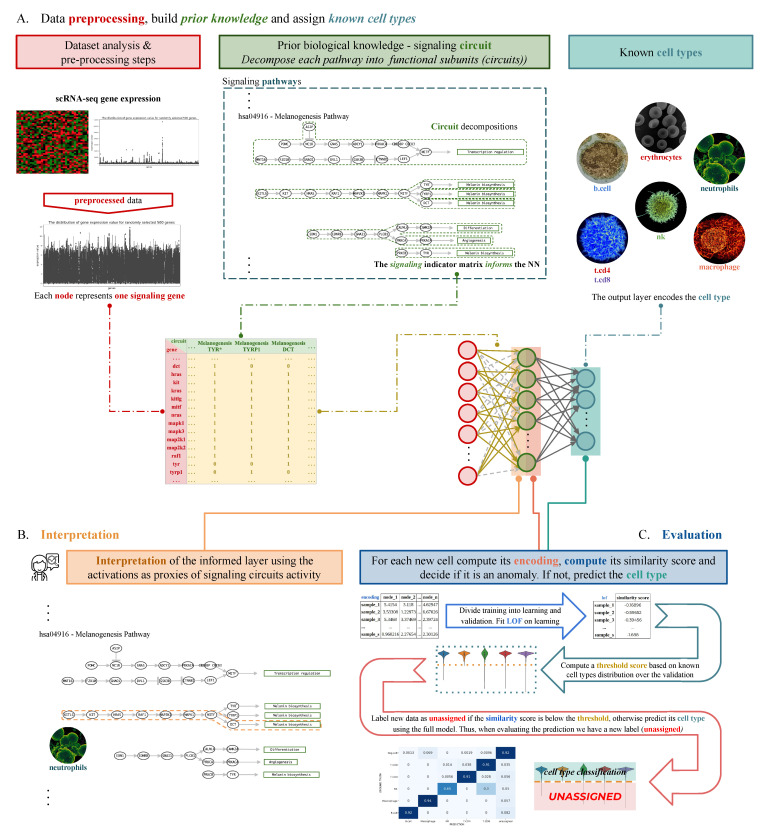
The first step (**A**) consists of preprocessing the data, building the signaling-informed layer *S*, and designing the architecture of the network based on the constraints imposed by an indicator matrix $I_{S}$ (see Methods). The second step (**B**) deals with the interpretation using the functional characterization of each cell cluster by aggregating the activations of the informed layer with respect to each observed or predicted cell type. The final step, (**C**) consists in making new predictions by (i) dividing the *training* set into *learning* and *validation*, (ii) fitting an anomaly detection algorithm to the encodings of the *learning* set, compute a threshold with *validation*, and (iii) label a new cell as unassigned if the threshold is not met, otherwise use the cell type prediction of the full NN.

**Figure 2 biology-12-00579-f002:**
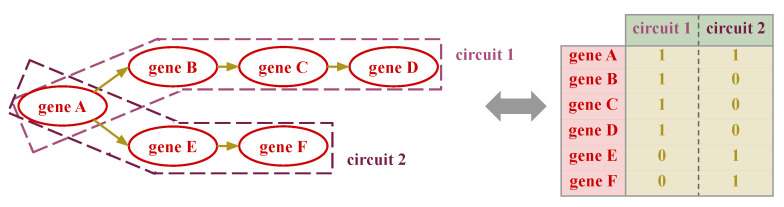
Simplified version on how to decompose a pathway into effector circuits and build the corresponding indicator matrix. On the left side, we see a simplified pathway that gives rise to two effector sub-pathways (referenced as *effector circuits* in this work), which lead to the indicator matrix depicted on the right side.

**Figure 3 biology-12-00579-f003:**
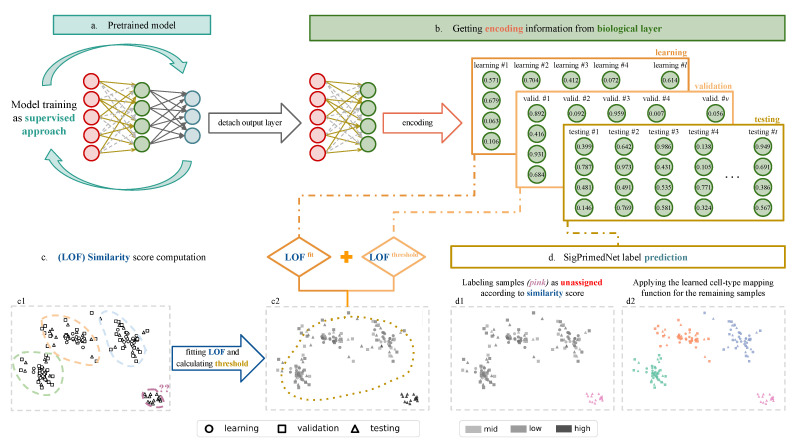
SigPrimedNet’s prediction mechanism (in two dimensions to simplify). Given the fitted NN (**a**), we detach the output layer and compute the encodings of the learning, validation and test sets (**b**), Then, in (**c1**) we fit a LOF to model the training encodings, (**c2**) predict the similarity scores of the learning and validation encodings to compute a more realistic threshold. Finally, (**d1**) a sample is labeled as unassigned if its similarity score is bellow the threshold, if not (**d2**) we apply the cell-type mapping learned by the NN.

**Figure 4 biology-12-00579-f004:**
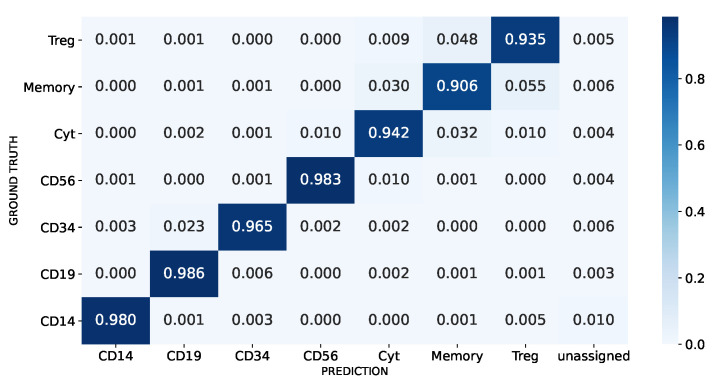
PBMC aggregated cross-validation confusion matrix. The unassigned label refers to cells that the model could not assign a known cell type.

**Figure 5 biology-12-00579-f005:**
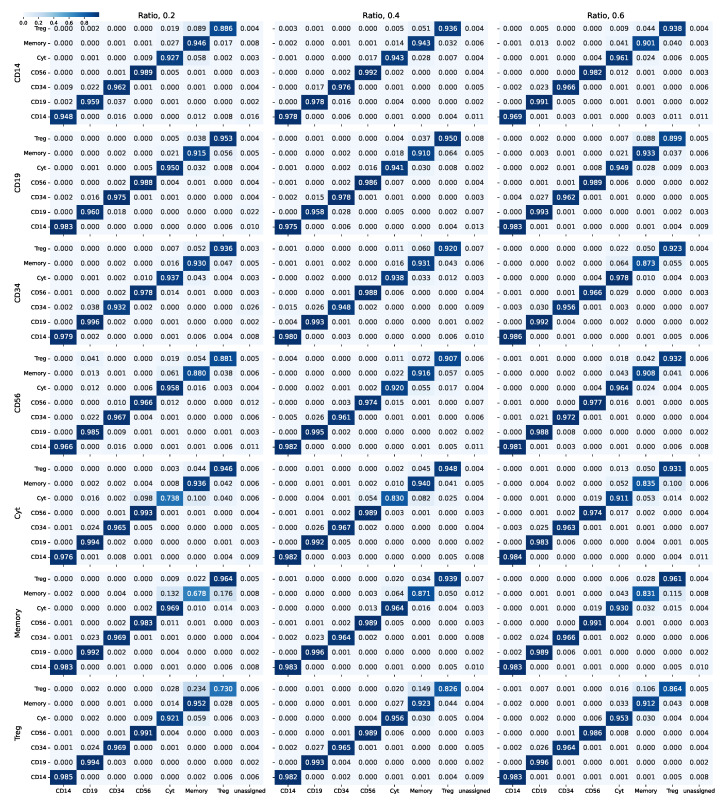
Summarized confusion matrices for the different unbalanced simulations using the PBMC dataset. The rows represent the cell type being undersampled, while the columns are the fraction of cells kept for each cell type.

**Figure 6 biology-12-00579-f006:**
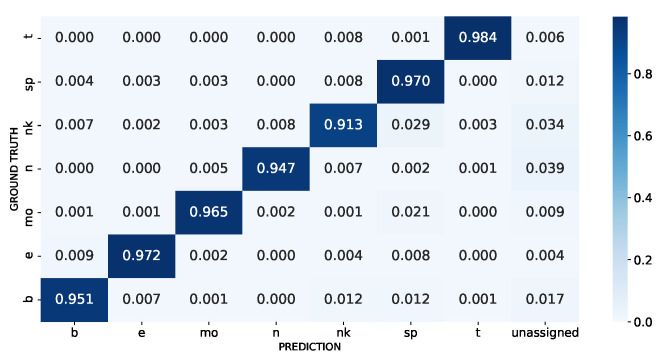
The aggregated cross-validation confusion matrix of the Immune dataset. The cell types b, e, mo, n, nk, sp, and t refer to B cells, erythrocytes, monocytes, neutrophils, NK cells, CD34+ HSPCs, and T cells, respectively. The unassigned label refers to cells that the model that could not correctly assign a known cell type.

**Figure 7 biology-12-00579-f007:**
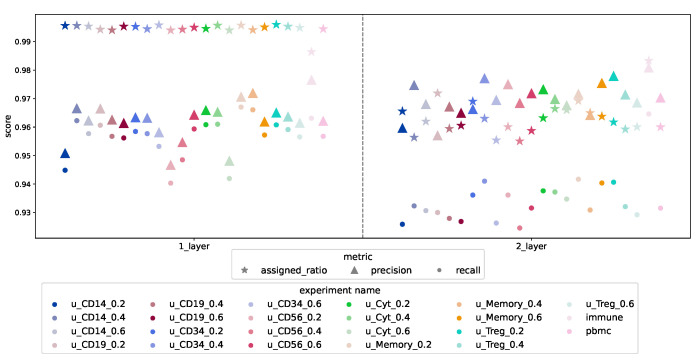
The colors represent the datasets used for testing the performance. In the legend “immune” refers to the Immune experiment, “pmbc” to the balanced PBMC experiment, and u_cell-type_ratio refers to the undersampled version of the PBMC where the “ratio” indicates the proportion of cells of sampled cells a given “cell-type”. Triangles and dots represent the mean across the test sets of the weighted precision and recall scores, respectively, while the crosses resent the mean of the proportion of cells assigned a cell type (since all cells are known, the higher the better).

**Figure 8 biology-12-00579-f008:**
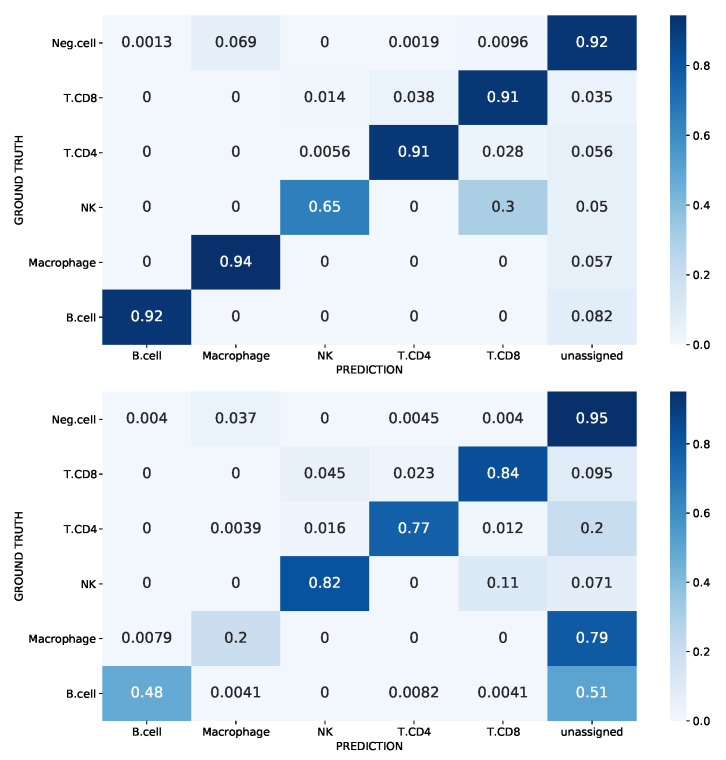
The confusion matrix of the melanoma dataset for the unknown cell-type identification task. SigPrimedNet (top) and Scibet (bottom).

**Figure 9 biology-12-00579-f009:**
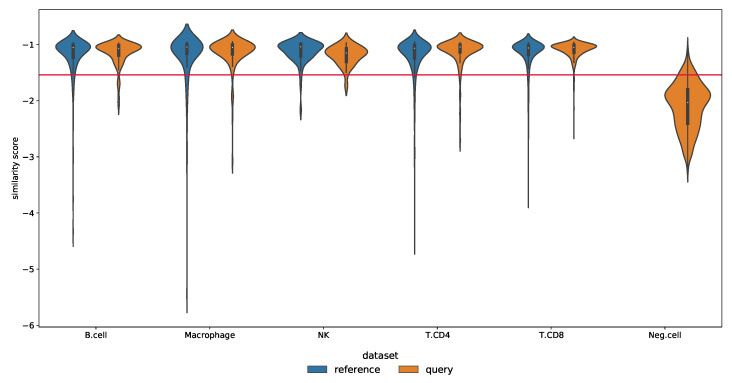
Similarity score distribution for each cell type on the validation and test splits (Melanoma dataset). The horizontal line shows the threshold obtained using the reference set inner splits as detailed in Section 2.6.

**Table 1 biology-12-00579-t001:** Cell type distribution for each dataset.

PBMC		Immune		Melanoma	
Cell Type	# of Samples	Cell Type	# of Samples	Cell Type	# of Samples
CD14+	2500	B cells	1465	B.cell	818
CD19+	2500	Erythrocytes	1747	Macrophage	420
CD34+	2500	HSPCs	3742	NK	92
CD56+	2500	Monocytes	954	T.CD4+	856
CD8+ Cytotoxic	2500	Neutrophils	485	T.CD8+	1759
CD4+/CD45RO+ Memory	2500	NK	546	Negative cells	2228
Treg	2500	T cells	517		

**Table 2 biology-12-00579-t002:** Hyperparameter values.

Dataset	Hyperparameter	Hyperparameter Value
PBMC	epochs	100
	batch_size	10
Immune	kernel_initializer	glorot_uniform + sig-informed
	bias_initializer	zeros
Melanoma	activation	relu (hidden layers)/softmax (last layer)
	optimizer	Adam

**Table 3 biology-12-00579-t003:** Execution times.

		DESIGN	
Dataset	Experiment	1-Layer	2-Layer
PBMC	RepeatedStratifiedKFold (10 k-fold with 50 iterations)	mean, 3.20 min std, 0.77 min total execution time is 13.28 h	mean, 3.26 min std, 0.87 min total execution time is 13.52 h
Immune	RepeatedStratifiedKFold (10 k-fold with 30 iterations)	mean, 2.82 min std, 0.69 min total execution time is 14.07 h	mean, 4.31 min std, 1.30 min total execution time is 21.48 h
	train_test_split (50% test size with 100 iterations)	mean, 1.94 min std, 0.92 min total execution time is 3.2 h	mean, 1.79 min std, 0.51 min total execution time is 2.95 h
Melanoma	training with reference dataset (one iteration)	total execution time is 1.96 min	total execution time is 3.7 min

**Table 4 biology-12-00579-t004:** Comparison between SigPrimedNet (one-layer (1L) and two-layer (2L) designs) and the best PDNN design on the Melanoma test split.

	MACRO			WEIGHTED				
Design	F1	Precision	Recall	F1	Precision	Recall	Accuracy	Balanced Accuracy
SigPrimedNet (1L)	0.838	0.823	0.884	0.926	0.945	0.919	0.919	0.884
SigPrimedNet (2L)	0.743	0.785	0.796	0.878	0.927	0.846	0.846	0.796
PDNN	0.861	0.922	0.844	0.933	0.938	0.936	0.936	0.844
PDNN (*)	0.499	0.454	0.753	0.241	0.224	0.326	0.326	0.753

## Data Availability

The full version of the PBMC dataset can be obtained from 10× Genomics [24], the Melanoma dataset can be obtained from GEO [23], and the Immune dataset can be obtained from [26]. Finally, the filtered versions used in this study can be obtained from [15] (PBMC and Melanoma) and [25] (Immune).

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
