# Peer review of "SigPrimedNet: A Signaling-Informed Neural Network for scRNA-seq Annotation of Known and Unknown Cell Types"

_biology, 2023, doi:10.3390/biology12040579_

Round 1

Reviewer 1 Report

Single cell sequencing technology has generated increasingly large dataset, and inspired numerous follow-up studies, but interpretation of the data has posed a hurdle for utilization of datasets. Constructing the link between the sequencing results and biological cell type/states is a key step in interpreting the sequencing data, but the current algorithms tend to be set back by complicated datasets. Here this study reported a neural network based annotation algorithm that robustly identified cell types from sequencing datasets and provided interpretable results.

The algorithm described here can be beneficial to all working on single cell sequencing, but some further validation would be required. In order to demonstrate the robustness of the algorithm in unbalanced dataset, it is important to show the performance in synthetic datasets, where the ratio of different cell types are artificially controlled and compare the performance of the algorithm in datasets of biased cell types. In addition, the time it took to finish model training and classification is also important, given the increasingly large datasets. It is important to show the model training and running time with datasets of different sizes (e.g. merged large datasets or downsized smaller datasets) to provide a reference for the overall scalability of the algorithm.

Author Response

We appreciate very much the constructive comments of the referee that have contributed to increasing the quality of the manuscript. See below the point-by-point responses to them:

COMMENT

=========

In order to demonstrate the robustness of the algorithm in unbalanced dataset, it is important to show the performance in synthetic datasets, where the ratio of different cell types are artificially controlled and compare the performance of the algorithm in datasets of biased cell types.

Answer

======

We agree with the reviewer that constructing controlled synthetic datasets is the usual procedure for validating a method against unbalanced data. However, these procedures usually rely on mimicking a global correlation structure, but we need a gene structure that resembles the biological patterns found in the signaling pathways, since they inform the construction of the network. Nevertheless, to test the reliability of the model on (controlled) unbalanced datasets, we have randomly undersampled each cell type (using a proportion of 0.2, 0.4, and 0.6 of the original population) to produce a total of 21 synthetic datasets derived from the PBMC dataset. Then, we have run a ten-fold cross-validation procedure using the full SigPrimedNet model (for both, 1 and 2-layer designs) for each synthetic dataset.

We have added a new subsection in the results section to accommodate the new results, which confirm the performance previously observed.

COMMENT

=========

In addition, the time it took to finish model training and classification is also important, given the increasingly large datasets. It is important to show the model training and running time with datasets of different sizes (e.g. merged large datasets or downsized smaller datasets) to provide a reference for the overall scalability of the algorithm.

ANSWER

=======

We recognize the value of having this information for the scientist willing to try our method in their data. We have moved the table to the main manuscript (new Table 3), which was previously buried in the appendices and unproperly linked.

As part of the revision, we have also checked the grammar, typos, etc.

Reviewer 2 Report

Gundogdu and colleagues presented a method called SignPriemedNet, which integrates signaling pathway knowledge in a neural network, for cell type classification in single-cell RNA-seq data. The idea of using prior knowledge in a neural network has been demonstrated by others, e.g. https://doi.org/10.1186/s13059-020-02100-5, and the authors https://doi.org/10.1186/s13040-021-00285-4.

In the current manuscript, the authors benchmarked their methods with a method called SciBet, which does not make use of a similar approach in cell type classification. But the authors didn't compare their current methods with their previous work as shown in https://doi.org/10.1186/s13040-021-00285-4, or the work by others, which may implement the similar idea. It is also found that the dataset used, figures and texts in the current manuscript are extensively overlapping with the authors' previous work (https://doi.org/10.1186/s13040-021-00285-4).  The methodology, including the description of the datasets and formulation of the neural network, in the authors' previous work is more detailed than the current one. I am not going to give further comment on the current manuscript because without detailed descriptions of the datasets used, the implementation of the method and the codes, it is hard to evaluate the current work. I have an impression that the current manuscript is largely repeating the previous one (https://doi.org/10.1186/s13040-021-00285-4) and the authors should make it very clear how the current method is different from the previous one. 

Author Response

We appreciate very much the constructive comments of the referee that have contributed to increasing the quality of the manuscript. 

We fully agree that the presentation of the manuscript poorly communicated the novelty and improvements over our previous manuscript. In order to improve these aspects, we have committed to a series of major revisions, including numerous new experiments.

Firstly, the main figure of the manuscript (Figure 1) has been completely overhauled to showcase the workings of our methodology, with an emphasis on facilitating the description of the inner workings of the model along the manuscript, and its ability to identify unknown cell types (an ability lacked by the referenced methods).

Secondly, guided by the structure of the new figure, we have added a more concise explanation of how we compose the structure of the Neural Network, especially so when dealing with the biologically informed layer. We have used a more consistent notation and nomenclature throughoutthe text, and we have greatly expanded the indicator matrix construction (a critical step) with a more precise description, which we support with graph-based procedures and a new figure depicting a toy example. In addition, we have included the equations of some key concepts and a summary table with the main design choices (previously buried in the appendices).

Thirdly, the previous version of the manuscript described the full SigPrimNet method but then we evaluated a reduced model which could only work under supervised scenarios, except for the Melanoma dataset, where unknown cell types required the full model ability to identify them. Therefore, we have moved all the supervised-based results to a new Supplementary file, as we think that they could be still useful for the research community when comparing our method to other methodologies that lack the ability to identify unknown cell types.

Fourth, we have re-run all the experiments that dealt with known cell types, namely the PBMC and Immune experiments, using the complete version of SigPrimedNet. As SigPrimedNet could potentially identify cells as unknown, this ends in a more consistent and realistic evaluation of the misclassification rates of the model. As can be seen in the new figures in the result section, the proportion of cells labeled as unassigned by SigPrimed net is very low even under such a hard premise (all cell types are known). Since these experiments were proposed and rigorously tested in SciBet and we use exactly the same data and validation procedure, our findings are directly comparable to the models already tested in the paper.

Fifth, as requested by Reviewer 1 we have conducted a new set of synthetic experiments to showcase the performance of the proposed model when dealing with unbalanced scenarios. To do so, we have undermined the cell populations of the PBMC dataset one cell type each time, using a prespecified array of undersampling ratios (0.2, 0.4, 0.6) totaling 21 datasets. We evaluated the model using a ten-fold cross-validation schema for each of the 21 datasets. The results reinforce the performance of the model shown in the originally unbalanced Immune dataset.

Sixth, we also re-run all the experiments for the complete version of SigPrimedNet using a two-layer design. As it turns out, the results are very similar between the two models, so we have focused on the more interpretable one-layer design for the main manuscript. However, we think that it is interesting that adding more layers does not improve the quality of the results in a meaningful way. Thus, we have included a new subsection and a summary figure where we compare the one and two-layer designs for precision, recall, and the assigned ratio scores across the full set of PBMC (balanced and unbalanced) and Immune experiments. As it can be still interesting, we moved the definition and results of the less interpretable two-layer design to the new supplementary file.

Finally, we have fixed all the typos and grammar inconsistencies that we found, and we have also simplified the appendices, making it easier to find the pathways used, what the model has found interesting, and the software used. To facilitate the review process, we have included numerous comments using the margins of the manuscript.

Round 2

Reviewer 2 Report

Gundogdu and colleagues presented a  revised manuscript describing the SignPriemedNet method for cell type classification based on neural network. While the authors have revised the figures and texts, there are still issues that the authors must address.

1. The authors has revised the methodology part to make it more comprehensive. It is getting more clear to me how the prior knowledge is processed and incorporated into the weight matrix between the input and first (hidden) layer. Yet, it is unclear how the number of nodes (cell types) in the output layer is determined? Is it soly based on the cell type labels in the training data? Is there always a node for the unassigned (negative) cell type? In line 255, the authors mentioned about "the learning encodings". Are these the value outputs from the hidden layer, e.g. in the one-layer model? How were the anomaly and similarity scores were calculated?

2. The authors have made the GitHub repository available in this manuscript. I notice that the core Python scripts in the scripts directory in https://github.com/babelomics/sigprimednet and https://github.com/babelomics/signalization_prior_knowledge_based_nn are exactly the same. Does it mean that the authors simply duplicated the codes from their previous study (https://doi.org/10.1186/s13040-021-00285-4)? If this is the case, there seems to be no novelty in this manuscript compared with what the authors have published. If there are indeed improvements in this manuscript compared with the previous one, shouldn't the authors do a side-by-side comparison between the two studies?

Based on the contents presented, I would not be able to see the originality and novelty of this manuscript, especially for the methodology part.

Author Response

We appreciate very much the time and efforts of the reviewer, whose constructive comments have contributed to increase the quality of the manuscript. See below the point-by-point responses to them.

Response to 1

===========

As the reviewer has pointed out, the model first fits a (supervised) NN using the existing cell types: (referenced as the “known cell types” across the manuscript). However, we do not use additional nodes to give space to “unknown cell types”. Instead, we use an unsupervised learning method (Local outlier factor, LOF) that is trained using the activations learned by the model, also referenced as “encodings” or “learned representation” in the manuscript, which are computed by detaching the output layer. The encodings, also known as meta-features in the literature, have enough information to represent the cell-type cluster structure. But, as it is often the case, we need to mitigate the overconfidence of the NN when dealing with samples seen during the training. Thus, we split the training into learning and validation sets. The NN is fitted on the learning set, whose encodings are used to fit an unsupervised local outlier factor model, then we compute the validation encodings and the corresponding similarity scores (using the previously fitted LOF model). Now we can set a more realistic threshold using the similarity scores of the validation encodings. Finally, when new data arrives (cells), we first compute the encodings and the associated LOF similarity score, those cells below the threshold are labeled as “unassigned”, whereas the cells above the threshold are labeled according to the NN output predictions (the known cell types). In order to make it clearer to the reader we have added another toy example (see the new Figure 3) and we have fixed the score nomenclature across the manuscript.

We hope these additions make our results clearer for the reader.

Response to 2

===========

We understand the confusion. We have several repositories containing the same information for administrative reasons. First, we need to have the repositories in the “official” github account of the laboratory (github/babelomics). Second, Peling Gundogdu, the first author of the manuscript under revision, has been part of the Marie Curie Innovative Training Network entitled “Machine Learning Frontiers in Precision Medicine” and she needed to have her own repository linked in the internal reports, etc. Third, a fork of this repository was created for the previous study (https://doi.org/10.1186/s13040-021-00285-4) in babelomics/signalization_prior_knowledge_based_nn. Finally, we created another fork for the current study in (github/babelomics/sigprimednet).

Pelin’s own repository has been continually updated to make it easy for the MLFPM reports. While the previous manuscript was being redacted, the novel approach of the current manuscript was already an ongoing project. Thus, when we forked it to make it available in babelomics/signalization_prior_knowledge_based_nn for the reviewers and the research community it already contained the first steps of SigPrimedNet, although it was never part of the previous study. Moving forward to the current manuscript's first revisions, Pelin updated her own repository, then babelomics/signalization_prior_knowledge_based_nn was synced, and babelomics/sigprimednet was updated. We have undone the babelomics/signalization_prior_knowledge_based_nn commits because the repository should be frozen in time. In order to avoid future mistakes when referencing the code base used in the current manuscript we have created a github release for it (https://github.com/babelomics/sigprimednet/releases/tag/v1.0.0).

We hope that these explanations help to clarify the issues with the repositories. We think that our method is novel since we have a compromise between supervised and unsupervised learning using a domain-informed NN, which learns representations that are easy to interpret by domain experts. Our previous work was unable to deal with unknown cells, thus the comparison is unrealistic. However, we have added a paragraph when presenting the results of the Melanoma dataset. Note that in the previous study, we removed the negative cells from the test split of the dataset because the method was unable to deal with “unknown” cell types. Even if the comparison is unfair (the filtering favors our previous method), the performance is on the same scale. However, it should be noted that if the previous method is used in the full dataset (with negative cells), all the “negative” cells would be classified using one of the known cell types. In our humble opinion, this limitation that we have overcome with SigPrimedNet is novel enough (to the best of our knowledge SigPrimedNet is the first domain-informed NN with such capabilities). Furthermore, SigPrimedNet makes use for the first time in a NN the effector sub-pathway decomposition, which could enable more precise interpretations (eg. https://doi.org/10.1093/narcan/zcaa011).